# Late-Onset Bleb-Related Endophthalmitis Caused by *Moraxella nonliquefaciens*: A Case Report

**DOI:** 10.3390/antibiotics12030607

**Published:** 2023-03-18

**Authors:** Su-Chin Shen, Kuan-Jen Chen

**Affiliations:** 1Department of Ophthalmology, Chang Gung Memorial Hospital, Taoyuan 333, Taiwan; 2College of Medicine, Chang Gung University, Taoyuan 333, Taiwan

**Keywords:** endophthalmitis, glaucoma filtering surgery, *Moraxella nonliquefaciens*

## Abstract

*Moraxella* species are Gram-negative coccobacilli that typically colonize the flora of the human upper respiratory tract and have low pathogenic potential. There are limited case reports implicating the organisms as the cause of endocarditis, bacteremia, septic arthritis, ocular infection, and meningitis. In cases of keratitis and conjunctivitis, *Moraxella nonliquefaciens* is not commonly isolated from the ocular surface. We present a case of a diabetic patient who developed late-onset bleb-related endophthalmitis caused by *M. nonliquefaciens* 4 years after glaucoma filtering surgery. Within one day, the patient presented with an acutely fulminant course with sudden visual loss, redness, and ocular pain. Appropriate antibiotic treatment and early vitrectomy resulted in a favorable final visual acuity of 20/100, which was his vision prior to infection. The use of Matrix-Assisted Laser Desorption Ionization–Time of Flight Mass spectrometry (MALDI-TOF MS) enabled the rapid identification of the organism. Endophthalmitis caused by *M. nonliquefaciens* should be considered in patients who underwent glaucoma filtering surgery with antifibrotic agents.

## 1. Introduction

Bacterial endophthalmitis, which can lead to significant loss of vision, is a concerning complication of ocular surgery. Most cases of endophthalmitis are exogenous and result from ocular surgical procedures, especially cataract surgery. The Endophthalmitis Vitrectomy Study defined acute-onset postoperative endophthalmitis as infections that occur within six weeks after the surgery, whereas delayed-onset postoperative endophthalmitis was defined as infections that occur after a period greater than six weeks following the surgery. In post-cataract patients, delayed-onset cases of endophthalmitis are mostly caused by fungal or indolent bacterial infections, such as Cutibacterium acnes. These cases typically present as a persistent low-grade inflammation in the anterior chamber. Patients present with decreased vision in the affected eye, and half of them also have eye pain, which is usually mild. However, the time courses for endophthalmitis following cataract surgery and filtration surgery are distinct from each other. Late-onset endophthalmitis associated with filtering bleb, which occurs in an eye that has remained stable for several months or years after surgery, is less common, but carries a high degree of risk [1]. It can develop many years after the procedure without warning and progress rapidly. Additionally, the organisms responsible for this condition are often more virulent than those typically associated with post-cataract endophthalmitis. The frequency of late-onset endophthalmitis after glaucoma surgery ranges from 0.1% to 1.5% [2]. *Streptococci* are the most common pathogens involved in bleb-related endophthalmitis; other major pathogens include *Staphylococcus aureus*, *Hemophilus influenzae*, *enterococci*, and *Moraxella catarrhalis* [3].

*Moraxella nonliquefaciens*, a Gram-negative coccobacillus, is part of the normal microbiota of the human respiratory tract and is considered to have low pathogenic potential [4]. Ocular infections due to *M. nonliquefaciens* are usually limited to eye surface and present as conjunctivitis, keratitis, and corneal abscess [5,6]. Herein, we report an unusual case of late-onset bleb-related endophthalmitis caused by *M. nonliquefaciens*.

## 2. Case Report

A 72-year-old diabetic male presented to our emergency department with a 1-day history of red eye and sudden visual loss in his left eye. He had a known history of advanced primary angle closure glaucoma, which was treated with bilateral laser iridectomy and bilateral cataract extraction with a posterior chamber intraocular lens implantation 10 years prior. Left eye trabeculectomy with mitomycin C was performed 4 years prior. His right eye had no light perception due to end-stage chronic angle closure glaucoma. Slit-lamp biomicroscopic examination of left eye showed opacified bleb surrounded by hyperemic conjunctiva and cornea edema. The filtering bleb was congested, and a 2 mm hypopyon and vitreous clouding were observed (Figure 1A). No fundal red reflex was observed. Intraocular pressure was 24 mm Hg and visual acuity decreased to light perception. The fundus was invisible and B-scan ultrasonography showed increased echogenicity in the vitreous cavity with no evidence of the posterior hyaloid detachment. Following aqueous and vitreous tapping, emergent pars plana vitrectomy was performed. Multiple yellow-white vitreous condensations were removed and the fundus showed diffuse retinal hemorrhages as well as retinal vasculitis. Intravitreal vancomycin (1 mg/0.1 mL), ceftazidime (2.25 mg/0.1 mL), and dexamethasone (0.4 mg/0.1 mL) were administrated at the end of surgery. Gram stains from vitreous revealed Gram-negative coccobacillus. Both bacterial cultures from aqueous and vitreous fluids showed *M. nonliquefaciens* growth, which was confirmed by Matrix-Assisted Laser Desorption Ionization–Time of Flight Mass spectrometry (MALDI-TOF MS). In vitro antibiotic susceptible testing showed that the *M. nonliquefaciens* isolate was sensitive to amoxicillin-clavulanic acid, cefuroxime, ceftriaxone, and trimethoprim-sulfamethoxazole. One month later, his visual acuity recovered to 20/200. The patient achieved a best-corrected visual acuity of 20/100, which was his vision prior to infection, during his 6-month follow-up (Figure 1B). Figure 2 illustrates his clinical progression.

## 3. Discussion

*Moraxella* species are oxidase-positive, aerobic Gram-negative cocci. They are easily identified by their classic appearance as brick-shaped diplobacilli, which are distinct from other Gram-negative bacteria. *Moraxella* spp. are a part of the normal flora of the human respiratory and genital tracts and are considered to have low pathogenic potential. Among *Moraxella* spp., the most clinically relevant microorganism from this genus is *M. catarrhalis*, which can cause otitis media in children and exacerbations of chronic obstructive pulmonary disease [7,8]. Other members of the *Moraxella* genus, such as *M. lacunata*, *M. osloensis*, and *M. nonliquefaciens* rarely cause disease.

Ocular infections due to Moraxella is low and usually limited to eye surface. The epidemiology of Moraxella keratitis has been studied in detail all over the world; this condition accounts for 2 to 3% of all cases of bacterial corneal ulcers between 2010 and 2018 [9,10,11]. Indeed, Moraxella may be considered an emerging pathogen, as a ninefold increase in the incidence of Moraxella keratitis was recently reported in Ireland [12]. A similar trend has also been observed in the USA and England, where Moraxella species appear to be on the rise, as an increase of 6% has been observed during the last decade [13]. The prevalence of *Moraxella* spp. in endophthalmitis is also increasing. In a large retrospective series at Bascom Palmer Eye Institute between 1991 and 2000, only 1.3% (10 of 757) of endophthalmitis cases revealed Moraxella species infection [14]. A subsequent French multicenter prospective study performed between 2004 and 2005 revealed that *Moraxella* spp. accounted for 6.5% (7 of 108) of the bacterial spectrum [15]. It is possible that these findings are indicative of alterations in the ocular microbiota profiles, which may be more accurately detected using new molecular diagnostic techniques that enhance the sensitivity and accuracy of pathogen identification.

*M. nonliquefaciens* is considered a part of the normal human flora of the upper respiratory tract, with few infections, such as osteomyelitis, meningitis, and endocarditis, being mentioned in the literature [16,17]. Ocular infections due to *M. nonliquefaciens* have rarely been characterized. The first report of ocular infection due to *M. Nonliquefaciens*, by Ebright et al., was concerned with a post-renal transplant, immunocompromised patient whose infection was believed to have been caused by minor trauma of the cornea from a contact lens [18]. Loube et al. reported two cases of *M. nonliquefaciens* endophthalmitis following trabeculectomy. Symptoms included pain, inflammation, and a rapid decrease in visual acuity [19]. A review of the literature shows only six other reported cases of late-onset bleb-related *M. nonliquefaciens* endophthalmitis and these infections occurred within a 2-month to 15-year period following surgery (Table 1) [1,16,19,20,21,22].

The majority of cases of late-onset endophthalmitis after glaucoma filtering surgery demonstrate a devastating visual outcome. The most common presenting complaints of bleb-related endophthalmitis include ocular pain (71%) and redness (53%), which typically appear within three days of presentation. Other common symptoms include blurred vision (35%), tearing (12%), purulent discharge (12%), and photophobia (10%). Findings include marked intraocular inflammation and, often, a bleb infiltrate [23]. *Staphylococcus* and *Streptococcus* species are the main pathogens. *Streptococcus* produces exotoxins causing a fulminant infection with poor visual outcome [24,25]. Other causes of bleb-related endophthalmitis are *H. influenzae*, *Acinetobacter*, and *M. catarrhalis*, among other bacteria. *M. nonliquefaciens* is generally considered to be of low virulence and is sensitive to most antibiotics.

A filtering bleb is a small sclera defect, created by glaucoma surgery, that allows excess aqueous humor to be absorbed into the systemic circulation. However, only a thin layer of conjunctiva separates the aqueous from the ocular surface in the location of the bleb, and this carries a risk of developing endophthalmitis. A large study from Japan reported the risk of endophthalmitis to be 1.3% per patient-year, and that it could occur between 1 month and 8 years following glaucoma surgery, with an average onset of 2 years. Leaking blebs increase endophthalmitis risk nearly fivefold [26]. Our patient presented with a thin opacified bleb with marked congestion, which is suspected to be an infection of the bleb preceding endophthalmitis. Several factors are known to be significantly associated with the development of infection following late-onset glaucoma surgery, with major factors including an inferiorly located bleb, the presence of bleb leakage, diabetes mellitus, and the use of antiproliferative agents, usually mitomycin or 5-fluorouracil (5-FU), concomitantly with surgery [25,27]. The most significant risk factor for endophthalmitis was found to be bleb leakage. Patients with a history of bleb leakage were 4.71 times more likely to develop an infection. According to Solus et al., bleb-related infections occurred more frequently in eyes with a limbus-based flap than in those with a fornix-based flap [28]. The initial stage of endophthalmitis development involves bacteria migrating across the filtering bleb. Therefore, patients must be educated to promptly report any signs of ocular surface infection, such as redness, pain, or discharge, to their clinic. Early detection of bleb infection can be facilitated through timely intervention. Bleb leakage must be repaired or treated as soon as it is detected.

In the past decade, antifibrotic agents such as mitomycin C and 5-FU have been increasingly used in glaucoma filtering surgery as they can help achieve a further decrease in intraocular pressure and improve the functional success of trabeculectomy with a high risk of surgical failure. Histologic studies have shown that the use of antifibrotic agents is associated with blebs that are less vascularized, have a thinner epithelium, and have more atrophic stroma. Because of the increased use of antifibrotic agents and the resulting thin, cystic, avascular blebs, there is growing concern about the increased risk of bleb-related endophthalmitis [29].

Identifying *Moraxella* species through routine laboratory techniques has been challenging. It is important to identify the species of *Moraxella*, as each species has a different antibiotic sensitivity. Many isolates cannot be identified by 16S rRNA sequencing alone [15]. With the use of a diagnostic tool, MALDI-TOF MS, we were able to rapidly identify the bacteria in our case. This ensured that our patient received appropriate therapy based on published antibiotic susceptibility data. A commercial system, Biolog GenIII (Hayward, CA, USA), can also be used to identify bacteria to genus and species [4,30]. Next-generation sequencing has been shown to be a promising technology for the identification of emerging pathogens and rare infectious diseases. These new sequencing technologies may enable the implementation of rapid and accurate genotypic drug susceptibility testing prior to the administration of antimicrobial therapy [31]. 

*M. nonliquefaciens* is not commonly implicated in human disease, so detailed susceptibility patterns are not fully known. Generally, *Moraxella* species are more susceptible to penicillin, ceftazidime (100%), cefazolin (98%), fluoroquinolones (100%) (ciprofloxacin, ofloxacin, and moxifloxacin), aminoglycosides (94–100%) (gentamicin, tobramycin, and amikacin), and polymyxin B (99–100%), but are less susceptible to trimethoprim (11%) [4]. A recent study reported increased resistance in clinical strains to macrolides [32]. *M. nonliquefaciens*, unlike *M. catarrhalis*, was found to be susceptible to vancomycin. This is the motivation for treating the present case with a direct intravitreal injection of vancomycin and ceftazidime, which resulted in a favorable outcome.

Endophthalmitis is a medical emergency as delayed treatment may result in permanent vision loss. The treatment strategy of endophthalmitis depends on the visual function of the patient and the extent of the inflammation. Surgical debridement of the vitreous humor is recommended for patients with fulminant endophthalmitis who present with severe vision loss or rapidly deteriorating vision. Early, prompt vitrectomy involves sample collection for microbiologic investigation and cleaning the vitreous cavity. Fortified, broad-spectrum antibiotics can be used not only at the end of vitrectomy but also during the surgery, and the anterior chamber and the vitreous cavity can also be washed with antibiotics. Vancomycin and ceftazidime or amikacin are the current drugs of choice. The injection is performed after a sample of vitreous humor is taken for Gram staining and culturation. A repeat intravitreal antibiotic injection may be administered based on culture results. Intravitreal steroid injection, which can reduce inflammation, is sometimes given as adjunctive therapy to preserve tissue integrity. Awareness of endophthalmitis risk in patients who have undergone glaucoma filtering surgery with antifibrotic agents is necessary. Prompt Gram staining is imperative for the early identification of the offending pathogen.

## Figures and Tables

**Figure 1 antibiotics-12-00607-f001:**
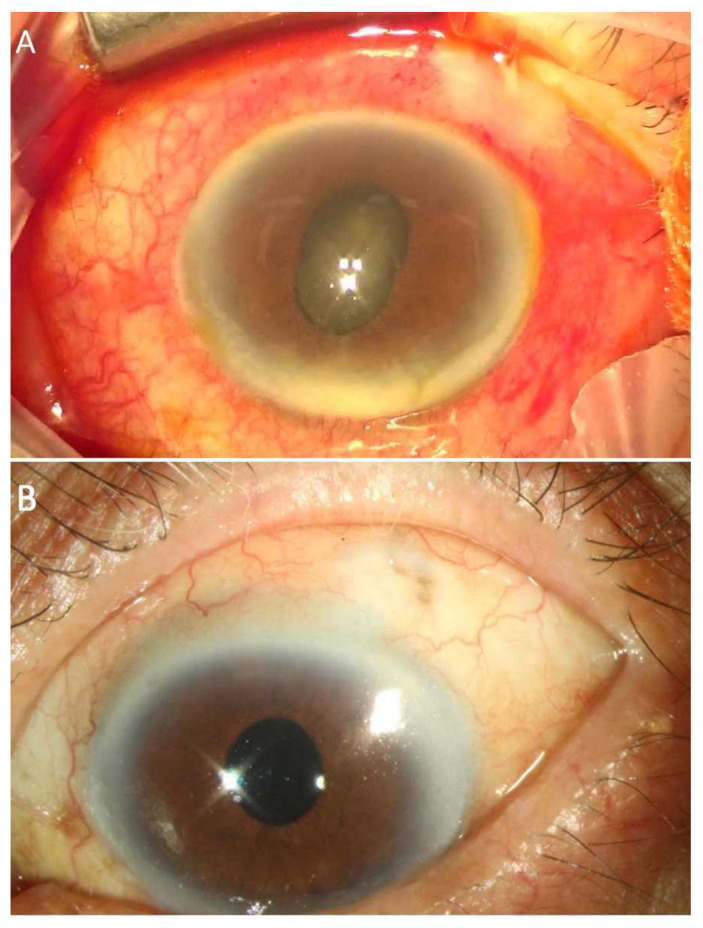
(**A**). Anterior segment revealed opacified bleb surrounded by hyperemic conjunctiva, corneal edema, hypopyon, and fibrinous reaction. (**B**). Six months later, anterior segment showed resolution of endophthalmitis without anatomical sequelae.

**Figure 2 antibiotics-12-00607-f002:**
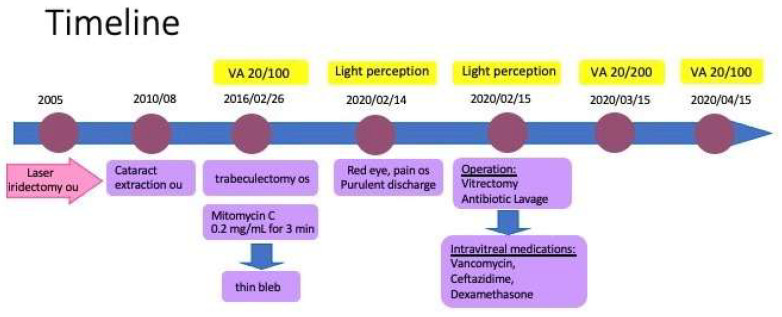
Clinical course.

**Table 1 antibiotics-12-00607-t001:** Trabeculectomy-related Endophthalmitis caused by *Moraxella nonliquefaciens* in literature and present case.

Patient	Author/ Year of Publication	Age/ Sex	Interval between Onset and Trabeculectomy/ Cataract Extraction	Underlying Disease	Initial VA	Management	Intravitreal Agents	Final VA
1	Lobue et al./1985 [19]	67/M	5 years/1 year		LP	Tap	GEN, CLI, D	20/50
2	Lobue et al./1985 [19]	62/F	15 months/No	DM	LP	Tap	GEN, CLD, D	HM
3	Mandelbaum/1985 [1]	73/M	15 years/No		20/200	Tap	GEN, CFZ	20/200
4	Sherman et al./1993 [20]	70/F	5 months/5 months		HM	Tap/PPV	CFZ, TOB	20/200
5	Schmidt et al./1993 [21]	79/M	2 months/2 months		LP	PPV (twice)	VAN, AMK/GEN, CRO	NLP
6	Laukeland et al./2002 [16]	78/M	9 years/5 years		LP	Tap	VAN, GEN	≦LP
7	Laukeland et al./2002 [16]	76/M	2 months/4 years		HM	Tap	VAN, GEN	≦HM
8	Díaz Barrón/2020 [22]	90/F	10 years/10 years	PR	HM	Tap	VAN, CAZ	20/200
9	Present study	72/M	5 years/7 years	DM	LP	PPV	VAN, CAZ, D	20/100

Abbreviation: AMK = amikacin; CAZ = ceftazidime; CFZ = cefazolin; CLI = clindamycin; CLD = cephaloridine; CRO = ceftriaxone; D = dexamethasone; DM = diabetes mellitus; GEN = gentamicin; HM = hand motions; LP = light perception; NLP = no light perception; PPV = pars plana vitrectomy; PR = polymyalgia rheumatica; TOB = tobramycin; VA = visual acuity; VAN = vancomycin.

## Data Availability

Not applicable.

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
