# Peer review of "Late-Onset Bleb-Related Endophthalmitis Caused by Moraxella nonliquefaciens: A Case Report"

_antibiotics, 2023, doi:10.3390/antibiotics12030607_

Round 1
Reviewer 1 Report
The article is well written. I believe that this paper can be accepted and published without revisions.
Author Response
Dear Reviewer,
We would like to express our appreciation for your generous comments.
Reviewer 2 Report
please find attachment

Author Response
Dear Reviewer,
We would like to express our appreciation for your valuable comments.
Valuable, interesting manuscript, precise literature review, useful information for every-day practice.
Line 27-28: Late onset endophthalmitis: What is the source of this definition? The time courses of post-cataract or post-fitration endophthalmitis are different. Please describe the main differences between post-cataract and post-filtration endophthalmitis.
Reply: We would like to express our gratitude to the reviewers for their comments. We have included the reference to Mandelbaum et al. (1985) in the manuscript to define late onset endophthalmitis as the type of infection that develops in an eye that has been stable for months or even years following a surgical procedure. Furthermore, we have added the following sentences to clarify the differences between post-cataract and post-filtration endophthalmitis. The Endophthalmitis Vitrectomy Study defined acute-onset postoperative endophthalmitis as infections that occur within six weeks after the surgery, whereas delayed-onset postoperative endophthalmitis was defined as infections that occur after a period greater than six weeks following the surgery. In post-cataract patients, delayed-onset cases of endophthalmitis are mostly caused by fungal or indolent bacterial infections, such as Cutibacterium acnes. These cases typically present as a persistent low-grade inflammation in the anterior chamber. Patients present with decreased vision in the affected eye, and half also have eye pain, which is usually mild. However, the time courses for endophthalmitis following cataract surgery and filtration surgery are distinct from each other. Late-onset endophthalmitis associated with filtering bleb, which occurs in an eye that has remained stable for several months or years after surgery, is less common but carries a high degree of risk. It can develop many years after the procedure without warning and progress rapidly. Additionally, the organisms responsible for this condition are often more virulent than those typically associated with post-cataract endophthalmitis.
Line 57: How did you check whether the posterior hyaloid was detached?
Reply: Indeed, B-scan ultrasonography was used to evaluate the patient's condition, and the absence of posterior hyaloid detachment was noted. This result has been added to the revised manuscript for clarity.
Line 105: Blebitis often precedes the endophthalmitis after filtrating surgery. (Lehmann OJ, Bunce C, Matheson MM, et al. Risk factors for development of post-trabeculectomy endophthalmitis. Br J Ophthalmol. 2000;84:1349 1353. PMID: 11090471)
It could be a very important threatening prodromal sign. Please add to Table 1 information about prior episodes of blebitis.
Reply: We would like to express our appreciation for the reviewer's comments. As per the literature, a bleb is a small defect in the sclera, and the prodromal signs of blebitis are typically mild and non-specific, such as tearing or increased eye discharge. However, the subsequent onset of endophthalmitis typically manifests acutely with intense pain, redness, and rapid visual loss of the infected eye, which often overwhelms the prodromal signs of blebitis. According to a literature review by Samra Waheed et al., the most common presenting complaints of bleb-related endophthalmitis include ocular pain (71%) and redness (53%), typically appearing within three days of presentation. Other common symptoms include blurred vision (35%), tearing (12%), purulent discharge (12%), and photophobia (10%). We have added this information to the discussion paragraph instead of including it in Table 1 for better clarity.
Please add to Discussion: how can we prevent the endophthalmitis especially after filtration procedure? Line 117 More comments regarding the risk factors and the management of the patient should be added in order to connect the clinical case to the existing literature.
Reply: We appreciate the reviewer's comments, and in response to their suggestion, we have included a detailed description for further clarification in the revised manuscript. “The most significant risk factor for endophthalmitis was found to be bleb leakage. Patients with a history of bleb leakage were 4.71 times more likely to develop an infection. According to Solus et al, bleb-related infections occurred more frequently in eyes with a limbus-based flap than in those with a fornix-based flap. The initial stage of endophthalmitis development involves bacteria migrating across the filtering bleb. Therefore, patients must be educated to promptly report any signs of ocular surface infection, such as redness, pain, or discharge, to their clinic. Early detection of bleb infection can be facilitated through timely intervention. Bleb leakage must be repaired or treated as soon as it is detected.”
Line156-167: As to EVS, the treatment strategy of endophthalmitis depends on the visual function and the extension of the inflammation. EVS is a very useful, but old study. A more aggressive approach is: ubi pus ibi evacua. The main goals of early, prompt vitrectomy are sample collection for microbiologic investigation and cleaning the vitreous cavity. Antibiotics can be used not only at the end of vitrectomy but during the surgery, the anterior chamber and the vitreous cavity can also be washed with antibiotics.
Reply: We would like to express our gratitude for the reviewer's comments. In response to their suggestion, we have reworded the final paragraph and added their comments to the revised manuscript for better clarity and accuracy. “The treatment strategy of endophthalmitis depends on the visual function and the extension of the inflammation. Surgical debridement of the vitreous humor is recommended for patients with fulminant endophthalmitis who present with severe vision loss or rapidly deteriorating vision. Early, prompt vitrectomy is sample collection for microbiologic investigation and cleaning the vitreous cavity. Antibiotics can be used not only at the end of vitrectomy but during the surgery, the anterior chamber and the vitreous cavity can also be washed with antibiotics. “
Once again, we thank you for your generous comments.

Reviewer 3 Report
The authors reported a case of a patient who developed late-onset bleb-related Endophthalmitis. However, there are some questions should be addressed.
1. I suggest to delete the ‘and review of the literature’ in the title.
2. It would be better to add a clinical timeline to make the case presentation more clear.
3. In ‘Discussion’, it would be better to add some discussion about your case.
Author Response
Dear Reviewer,
We would like to express our appreciation for the valuable comments.
- I suggest to delete the ‘and review of the literature’ in the title.
Reply: As the reviewer concerned, manuscript title is revised.
- It would be better to add a clinical timeline to make the case presentation more clear.
Reply: As requested by the reviewer, the revised manuscript now includes an updated timeline.
- In ‘Discussion’, it would be better to add some discussion about your case
Reply: This case highlights the importance of promptly identifying and effectively treating eye infections to prevent potential vision loss. To further emphasize this point, we have included additional discussion in the last paragraph of the manuscript.è¡¨å–®çš„é ‚ç«¯è¡¨å–®çš„åº•éƒ¨
“The treatment strategy of endophthalmitis depends on the visual function and the extension of the inflammation. Surgical debridement of the vitreous humor is recommended for patients with fulminant endophthalmitis who present with severe vision loss or rapidly deteriorating vision. Early, prompt vitrectomy is sample collection for microbiologic investigation and cleaning the vitreous cavity. Fortified, broad-spectrum antibiotics can be used not only at the end of vitrectomy but during the surgery, the anterior chamber and the vitreous cavity can also be washed with antibiotics.”
Once again, we thank you for your generous comments.
